# Factors Associated with Burnout among Resident Physicians Responding to the COVID-19 Pandemic: A 2-Month Longitudinal Observation Study

**DOI:** 10.3390/ijerph19159714

**Published:** 2022-08-07

**Authors:** Teressa R. Ju, Emilia E. Mikrut, Alexandra Spinelli, Anne-Marie Romain, Elizabeth Brondolo, Varuna Sundaram, Cynthia X. Pan

**Affiliations:** 1Department of Medicine, NewYork-Presbyterian Queens, 56-45 Main Street, Flushing, NY 11354, USA; 2Department of Psychology, St. John’s University, 152-11 Union Turnpike, Jamaica, NY 11367, USA; 3Department of Surgery, Division of Vascular and Endovascular Surgery, Weill Cornell Medical College, 1300 York Avenue, New York, NY 10065, USA; 4Department of Surgery, NewYork-Presbyterian Queens, 56-45 Main Street, Flushing, NY 11354, USA; 5Department of Medicine, Weill Cornell Medical College, 1300 York Avenue, New York, NY 10065, USA; 6Division of Geriatrics and Palliative Care, NewYork-Presbyterian Queens, 56-45 Main Street, Flushing, NY 11354, USA

**Keywords:** graduate medical education, resident burnout, resident wellness

## Abstract

Background: Burnout during residency may be a function of intense professional demands and poor work/life balance. With the onset of the COVID-19 pandemic, NYC hospital systems were quickly overwhelmed, and trainees were required to perform beyond the usual clinical duties with less supervision and limited education. Objective: The present longitudinal study examined the effects of COVID-19 caseload over time on burnout experienced by resident physicians and explored the effects of demographic characteristics and organizational and personal factors as predictors of burnout severity. Methods: This study employed a prospective design with repeated measurements from April 2020 to June 2020. Participants were surveyed about their well-being every 5 days. Predictors included caseload, sociodemographic variables, self-efficacy, hospital support, perceived professional development, meaning in work, and postgraduate training level. Results: In total, 54 resident physicians were recruited, of whom 50% reported burnout on initial assessment. Periods of higher caseload were associated with higher burnout. PGY-3 residents reported more burnout initially but appeared to recover faster compared to PGY-1 residents. Examined individually, higher self-efficacy, professional development, meaningful work, and hospital support were associated with lower burnout. When all four predictors were entered simultaneously, only self-efficacy was associated with burnout. However, professional development, meaningful work, and hospital support were associated with self-efficacy. Conclusion: Burnout among residency is prevalent and may have implications for burnout during later stages of a physician’s career. Self-efficacy is associated with lower burnout and interventions to increase self-efficacy and the interpersonal factors that promote self-efficacy may improve physician physical and emotional well-being.

## 1. Introduction

COVID-19 is an infectious disease caused by a novel coronavirus, now known as Severe Acute Respiratory Syndrome Coronavirus-2 [1]. Since December of 2019 when the first case was reported, there have been a total of 539 million reported cases, and of those, 6.32 million patients died [2]. In March 2020, New York City (NYC) became the first US epicenter of the COVID-19 pandemic. To meet heightened clinical demands, resident physicians stepped into more independent roles with less supervision or support than is typical.

During public health crises, increased caseload and working hours can contribute to professional burnout, a psychological state characterized by physical/emotional exhaustion, feelings of cynicism, detachment from the workplace, and a depleted sense of professional accomplishment [3,4,5,6]. The literature has documented that burnout is prevalent amongst residents and other physicians in training prior to the pandemic, and the need for intervention and support have been well documented [7,8]. Factors that may buffer burnout among physicians include increased organizational support (e.g., mentorship, collegial support, and departmental guidance) [9,10,11] and personal resources such as self-efficacy, perceived meaning in work, and perceived professional development [12]. In the context of the COVID-19 pandemic, low social support and low self-efficacy have been associated with high burnout among residents [13]. However, much of the research on resident burnout during the COVID-19 pandemic employed cross-sectional designs that noted that the pandemic negatively affected resiliency and increased emotional distress [14,15]. Prior to the pandemic, studies evaluating factors associated with burnout and well-being amongst residents suggested self-efficacy is a salient predictor of burnout. To our knowledge, researchers have not employed longitudinal designs to examine the impact of the pandemic on burnout over time.

The present study examined the effects of COVID-19 caseload burnout experienced by residents responding to the pandemic over time and explored the effects of demographic characteristics, organizational factors, and personal factors as predictors of burnout severity.

## 2. Materials and Methods

### 2.1. Methods

The present study was conducted from April 2020 to June 2020 at NewYork-Presbyterian Queens (NYP/Q) to assess the epidemiology and risk factors of burnout in residents/fellows who provided direct care to patients with COVID-19. NYP/Q is a 535-bed tertiary community teaching hospital. Proposal was reviewed and approved by the Institutional Review Boards of NYP/Queens (IRB # 12820320).

### 2.2. Participants

All 183 residents/fellows at NYP-Queens with direct contact and care of COVID-19 patients were invited to participate in this study. Resident and fellow trainees were defined as medical school graduates who were receiving medical training within the Accreditation Council for Graduate Medical Education recognized programs. The investigators contacted the program directors for each training program and asked them to support the study. The email addresses of residents/fellows were provided by each program and study consents were delivered through email. The Qualtrics platform, an online survey platform, was used to email surveys to residents/fellows from 14 April to 31 June 2020. Participation was voluntary.

### 2.3. Study Designs and Survey Instruments

This study employed a longitudinal ecological momentary assessment strategy (EMA) [16] in which participants were surveyed about their well-being every five days as the pandemic response progressed. An intensive longitudinal approach using EMA permits the collection of data on key variables in real time, avoids problems with retrospective recall, and permits reliable changes over time in the phenomena of interest [17]. The duration of five days between surveys was chosen to prevent the surveys from being distributed on the same day, as hospital workload varies by day of the week [18].

The survey instrument included 18 items (see Appendix A) reflecting the systems approach that evaluates hospital, department, and clinician level factors advocated by the National Academy of Sciences [19]. Seven items assessing demographic and professional information (i.e., gender, age, race, marital status, professional role, years of experience, and department) were administered on the initial survey. Age was assessed using four category variables. Participants were asked how they were feeling on that day, and all items were rated on a five-point Likert scale indicating how much they agreed or disagreed with the statement indicating the feeling (with zero equaling strongly disagree and five equaling strongly agree). Ten items addressing clinical responsibilities, personal and professional resources, and burnout were administered at every assessment point. Evaluation of personal resources included a self-efficacy assessment of four items (Q10 to Q13), which were previously validated (New York-Presbyterian Queens self-efficacy scale, Cronbach’s alpha of 0.79) [20]. Two items (Q15 and Q16) assessed crisis-related professional growth and perceptions of meaningful work. One item (Q14) inquired about perceptions of the provision of timely information and support from the hospital. Burnout was assessed by a single-item validated measure of burnout in healthcare workers based on Dolan et al. (2015) [21]. Responses ranged from no burnout (score of one) to potentially debilitating burnout (scores of greater than two). Participants were regarded as reporting burnout if their scores were two or above, according to established criteria [11,22].

### 2.4. Data Collection and Statitical Analysis

This study employed a longitudinal ecological momentary assessment strategy (EMA) [16] in which participants were surveyed and the comprehensive survey reviews were conducted independently by two Ph.D. students in the research team who did not participate the study. Demographic data, such as age, gender, ethnicity, PGY year, and training department were collected. Age was collected in a range to protect confidentiality. Categorical variables were analyzed with a chi-squared (*x*^2^) test. Training level differences in categorical variables were analyzed with the chi-squared test, and training level differences were evaluated with analyses of variance at Time 1.

To evaluate gender or training level differences in burnout score, an analysis of variance (ANOVA) test was employed. A two-sided *p*-value of 0.05 or lower was considered statistically significant. To examine longitudinal changes over time in burnout, and the personal and professional resources associated with burnout, we performed multilevel mixed logistic regression analyses (MMLM) using the GLIMMPIX procedure in SAS software (version 9.4; SAS Institute Inc, Cary, North Carolina, USA). MMLM were employed in these analyses as these models are more robust to missing values than traditional regression analyses (Schafer and Yucel, 2002). In the analyses, time was treated as a continuous variable with a one-point change corresponding to a period of 5 days. For example, when 14 April 2020 was coded as 0, 15 April 2020 was then coded as 0.2, and subsequently, 20 April 2020 was coded as 0.4.

## 3. Results

Of 183 residents, 54 (30%) completed the survey. The rest of the residents/fellows did not complete the surveys due to various reasons, such as being actively infected with COVID-19, unwillingness to participate in the study, or not responding to the sent emails.

Table 1 demonstrated the demographic characteristics of participants. On average, residents completed 4.5 surveys with a range of (1–13). In total, 52% of participants were female in gender. The three highest-represented racial/ethnic groups were white (*n* = 18; 36%), Chinese (*n* = 7; 14%), and Korean (*n* = 7; 14%). With regard to PGY year, there were 13 PGY-1, 19 PGY-2, and 18 PGY-3 residents, respectively. Data from PGY-4 (*n* = 3) and PGY-5 (*n* = 1) resident participants were omitted from analysis due to concerns of small numbers of participants resulting in a breach of confidentiality. Chi-squared tests revealed no differences in gender by training year. In terms of training department, most resident physicians were in internal medicine (*n* = 27; 54%), followed by emergency medicine (*n* = 9; 18%), and surgery (*n* = 7; 14%).

### 3.1. Demographic Predictors of Burnout

At initial assessment, the mean burnout score was 2.62 (SD = 0.99), with 50% of participants exceeding the established criteria for burnout (i.e., scores greater than two). Women (M = 1.93) reported significantly more burnout than men (M = 1.36; F(1,47) = 4.19, *p* < 0.05). There were no differences in burnout by racial/ethnic identity.

There was a significant difference in burnout by training level (F(2,46) = 4.23, *p* < 0.05). PGY-3 residents (M = 2.18) reported more burnout than PGY-1 residents (M = 1.23; *p* < 0.01) and PGY-2 residents (M = 1.53; *p* < 0.05). PGY-1 and PGY-2 residents did not significantly differ. Data from PGY-4 (*n* = 3) and PGY-5 (*n* = 1) participants were omitted from these analyses due to concerns regarding confidentiality.

### 3.2. Burnout over Time

Two mixed model regression analyses indicated there was a significant negative effect of time on burnout (b = −0.05, *t*(136) = −2.83, *p* < 0.01) such that burnout decreased over time, even when controlling for caseload and gender.

Mixed models regression analyses revealed a significant interaction of time by training level on burnout (F(2124) = 3.90, *p* < 0.05). Over time, PGY-3 residents recovered faster compared to PGY-1 residents but did not differ from PGY-2 residents (Figure 1).

### 3.3. Personal and Professional Resources as Predictors of Burnout

To identify personal and professional resources associated with burnout, four mixed model analyses controlling for assessment time, caseload, gender, and training level were conducted. All personal and professional resources, including self-efficacy (b = −0.54, SE = 0.09, *t* = −6.02), perceived professional development (b = −0.21, SE = 0.09, *t* = −2.35), meaningful work (b = −0.29, SE = 0.07, *t* = −4.37), and hospital support (b = −0.28, SE = 0.08, *t* = −3.65) were individually negatively associated with burnout (all *p* < 0.05). When all resources were entered into the model, only the effects of self-efficacy remained significant (b = −0.60, SE = 0.07, *t* = −8.79, *p* < 0.0001, 95% CI: −0.73, −0.47). However, further analyses indicated that the other professional and personal resources were associated with self-efficacy (i.e., hospital support (b = 0.32, SE = 0.05, *t* = 6.54, *p* < 0.0001, 95% CI: 0.23, 0.42); meaningful work (b = 0.20, SE = 0.06, *t* = 3.28, *p* < 0.001, 95% CI: 0.08, 0.31); and professional development (b = 0.12, SE = 0.05, *t* = 2.18, *p* < 0.05, 95% CI: 0.01, 0.23)).

## 4. Discussion

This mechanistic and prospective study evaluated the trajectory of burnout among resident physicians responding to the initial phase of the COVID-19 pandemic. Notably, one in two residents reported burnout. Compared to estimates of burnout experienced during non-pandemic times, these findings indicate that the pandemic had culminated in psychological strain among postgraduate trainees [23]. The year of residency conferred different levels of risk at different points in time. Though PGY-3 residents reported the most burnout initially in April 2020, their scores decreased more quickly as caseload decreased as compared to PGY-1 residents. As more advanced trainees, PGY-3 residents may be the first group of trainees who assume responsibilities of attending physicians during increased clinical demand.

Burnout risk and recovery is a complex process informed by multiple personal and contextual factors. Consistent with prior literature, our results illustrate that self-efficacy, one’s perceived capability in handling challenges and taking care of oneself, is a salient predictor of physician burnout [24]. In the past, factors associated with resident burnout has been evaluated extensively across specialty and year of training [7,8,25,26]. A prior study evaluating self-efficacy amongst resident physicians noted that enhancing self-efficacy may help buffer the emotional exhaustion and communication of stress that is commonly associated with burnout [27]. Additionally, in a study of surgical residents, increased self-efficacy was also associated with lower burnout scores [28]. During the COVID-19 pandemic, there have been studies evaluating burnout amongst resident physicians; however, the majority have been cross-sectional studies, which did not follow the residents over time [14,15,28]. This present study took place in the early stages of the COVID-19 pandemic which may have posed serious threats to self-efficacy due to inherent difficulties treating a novel condition with undetermined treatment guidelines.

Residency is a time when physicians develop both professional skills and an awareness of their competence, leading to self-efficacy. At the same time, residents must also learn to monitor and manage their own health and, consequently, develop self-efficacy concerning these abilities. In this study, self-efficacy about professional development and self-care appears to be influenced by contextual factors within the professional space, such as hospital support and opportunities for professional development. Mentorship may be particularly important in fostering or sustaining self-efficacy, allowing greater opportunities for postgraduate trainees to observe tenured physicians model skills [29]. Despite time constraints, organizational interventions providing standardized mentorship may be critical to alleviating psychological burden among postgraduate trainees throughout public health crises. Mentorship interventions would be consistent with calls from the National Academies of Sciences to prioritize research systematically addressing physician well-being [13] and are strongly indicated for this already-vulnerable group [24,29].

This study is the first study to use a longitudinal ecological momentary assessment in resident physicians to evaluate burnout over time. The EMA strategy involved assessing residents’ burnout and resources repeatedly in real time over the first four months of the COVID-19 pandemic. The repeated measures allow for the evaluation of within-participant changes in burnout over time in context. In contrast to the single snapshot view provided by cross-sectional studies of the burnout experiences of residents, this longitudinal study provides a more reliable and intensive view of changes over time as the pandemic progressed. Half of the sample in this present study had three or more responses over the course of the study.

To note, there are several limitations associated with this study. First, the use of convenience sampling may have contributed to sampling bias. Second, the sample size was relatively small, and completion rate decreased over time. It is unclear in intensive longitudinal models how the rates of compliance or noncompliance affect the statistical power of the study. Future studies using intensive longitudinal modeling and the EMA approach should investigate this issue. Finally, the study was conducted at a single site during the initial height of the COVID-19 pandemic in Queens, New York City. As such, findings may not generalize to resident physicians in other geographic areas with differing patterns of COVID-19 infection, hospital caseload, and mortality rates.

Further research is needed to understand the trajectories of self-efficacy among resident physicians, as this mechanistic understanding can target interventions to reduce burnout. Further research can examine how hospital support and target professional development activities can help support physician self-efficacy in the face of a pandemic.

## 5. Conclusions

Resident physicians responding to the COVID-19 pandemic may experience differential risks for burnout over time. Self-efficacy appears to be a key predictor of burnout, but other factors including hospital support are associated with self-efficacy. Training mentorship which focuses on self-efficacy during times of crises may be especially valuable in the prevention of burnout.

## Figures and Tables

**Figure 1 ijerph-19-09714-f001:**
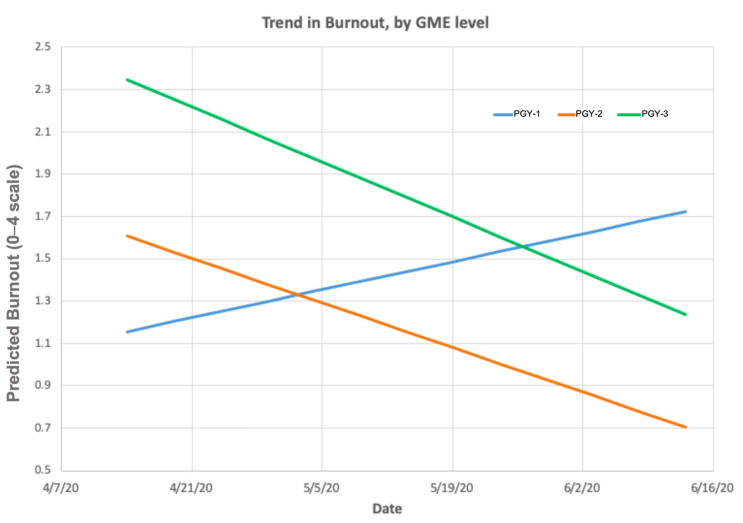
Trajectories of Burnout over time among PGY-1, PGY-2, and PGY-3 Resident Physicians.

**Table 1 ijerph-19-09714-t001:** Demographic Characteristics of the 54 participants.

	*N* (% of 54)
**Age**	
Under 35 years old	48 (88.9%)
35 or older	6 (11.1%)
**Gender**	
Women	28 (51.85%)
Men	26 (48.15%)
**Ethnicity**	
White	20 (37.04%)
Chinese	9 (16.67%)
Indian	6 (11.11%)
Alaska Native/American Indian	3 (5.56%)
Korean	7 (12.96%)
Other Asian	5 (9.26%)
Filipino	1 (1.85%)
Japanese	1 (1.15%)
Other	1 (1.85%)
Latino	1 (1.85%)
**PGY year**	
PGY-1	13 (24.07%)
PGY-2	19 (35.19%)
PGY-3	18 (33.33%)
PGY-4	1 (1.85%)
PGY-5	3 (5.56%)
**Department**	
Internal medicine	27 (50%)
Surgery	10 (18.52%)
Emergency Medicine	9 (16.67%)
Other	8 (14.81%)

## Data Availability

The data that support the findings of this study are available from the corresponding author upon reasonable request.

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
