# Peer review of "Factors Associated with Burnout among Resident Physicians Responding to the COVID-19 Pandemic: A 2-Month Longitudinal Observation Study"

_ijerph, 2022, doi:10.3390/ijerph19159714_

Round 1
Reviewer 1 Report
The manuscript has improved and clarified many of the points that were necessary to understand the methodology and phenomenon under study.
Just indicate that: Nothing relevant to the phenomenon under study has been provided in the introduction. It was indicated that studies that had analyzed the emotional impact of the pandemic on health professionals should be included.
Author Response
Reviewer:
Just indicate that: Nothing relevant to the phenomenon under study has been provided in the introduction. It was indicated that studies that had analyzed the emotional impact of the pandemic on health professionals should be included.
Reply:
Thank you for the suggestion. Within the introduction, we have included content referencing prior studies and their results on resident physician well-being during the pandemic.
This manuscript is a resubmission of an earlier submission. The following is a list of the peer review reports and author responses from that submission.
Round 1
Reviewer 1 Report
-Title: It would be necessary to indicate the observation period and identify the type of study. For example observational study.
-Introduction: A contextualization of the problem with an international character is recommended, before focusing specifically on the public health problem caused in New York.
-Methods: You must be clearer with the selection sample of the study. How it was carried out, the number, place, duration of the survey and questions to be evaluated. You must identify which scale was used in data collection and its validity.
It is necessary to identify the validation code of the study by the Ethics Committee.
The analytical plan section is not understood. It is necessary to incorporate statistical analysis and explain the tools used in it.
-Results:
The participating sample is from the methods section, not the results section.
Quantitative variables must be provided with a numerical value and as a continuous quantitative variable. Express the values with their mean and standard deviation, for example, age.
It is difficult to understand the importance and meaning of the mean values of the results if they do not first explain the possible scores achievable in the assessment survey.
There are variables that you do not define in the material and methods such as the level of training. When it identifies results associated with that variable, it cannot be logically correlated.
They should show the different analysis subgroups of their sample in the material and methods section. I am referring to the groups of residents per year and how many students make up those groups. He exposes it in table 1 but these are not results. They may not be representative as they have such a small sample of 54 individuals.
They must explain why 129 residents did not participate in the study. It has an excessively high percentage of non-participants and therefore the bias is too high in its results and conclusions.
In line 98, you use a logistic regression method that is not considered in the materials and methods section. It should preferentially address the statistical analysis used in this article.
Figure 1 illustrates the trend of burnout among residents of 1, 2 and 3 years, it does not include the entire sample due to the scarcity of study participants. The data is skewed in that illustration and no predictive values can be extracted.
-Conclusion: The phrase in line 163 is not proven by the results of your study.
In general, it must address a significant change in the methodology of the study carried out. One of the main drawbacks is the small sample used and the low participation of the total number of residents of the hospital.
Reviewer 2 Report
Some comments are suggested:
- The abstract must include which validated scale was used to measure burnout.
- It would be interesting if the keywords are DeCS/MeSH descriptors.
- The introduction needs to be expanded. Previous studies on burnout in COVID-19 and the effects on the emotional impact of professionals should be included. It should be described what important results have been obtained in other studies on demographic characteristics, organizational factors, and personal factors as predictors of burnout severity.
- In methodology, it is indicated that an ecological study has been carried out, but that implies using aggregate data and not individual data so as not to fall into an ecological fallacy. In this case, an important methodological bias has been incurred.
- The sample calculation or what sampling was used has not been explained. The measuring instruments have not been adequately explained. It is indicated that a validated scale has been used to measure burnout, but neither the validation data nor the reference to it are indicated.
- Statistical analysis is incompletely explained. Reference is made to mixed variance models, but which ones have you used? What statistical tests have you used for the prediction model, association, relationship between the variables, etc?
- The discussion should be improved since the results should be compared with similar studies, expanding the references to said studies.
- The conclusions must include some implications for practice and future lines or strategies of action must be proposed.